# A Synthetic Angle Normalization Model of Vegetation Canopy Reflectance for Geostationary Satellite Remote Sensing Data

Yinghao Lin [1,2,3], Qingjiu Tian [2], Baojun Qiao [1,*], Yu Wu [4,*], Xianyu Zuo [1], Yi Xie [1] and Yang Lian [5]

1   Henan Key Laboratory of Big Data Analysis and Processing, School of Computer and Information Engineering, Henan University, Kaifeng 475001, China
2   International Institute for Earth System Science, School of Geographic and Oceanographic Sciences, Nanjing University, Nanjing 210023, China
3   Zhong Ke Langfang Institute of Spatial Information Applications, Langfang 065000, China
4   School of Earth System Science, Tianjin University, Tianjin 300072, China
5   Henan Yellow River Administration Bureau, Zhengzhou 450053, China
*   Correspondence: qbj@henu.edu.cn (B.Q.); wu_yu@tju.edu.cn (Y.W.)

**Abstract:** High-frequency imaging characteristics allow a geostationary satellite (GSS) to capture the diurnal variation in vegetation canopy reflectance spectra, which is of very important practical significance for monitoring vegetation via remote sensing (RS). However, the observation angle and solar angle of high-frequency GSS RS data usually differ, and the differences in bidirectional reflectance from the reflectance spectra of the vegetation canopy are significant, which makes it necessary to normalize angles for GSS RS data. The BRDF (Bidirectional Reflectance Distribution Function) prototype library is effective for the angle normalization of RS data. However, its spatiotemporal applicability and error propagation are currently unclear. To resolve this problem, we herein propose a synthetic angle normalization model (SANM) for RS vegetation canopy reflectance; this model exploits the GSS imaging characteristics, whereby each pixel has a fixed observation angle. The established model references a topographic correction method for vegetation canopies based on path-length correction, solar zenith angle normalization, and the Minnaert model. It also considers the characteristics of diurnal variations in vegetation canopy reflectance spectra by setting the time window. Experiments were carried out on the eight Geostationary Ocean Color Imager (GOCI) images obtained on 22 April 2015 to validate the performance of the proposed SANM. The results show that SANM significantly improves the phase-to-phase correlation of the GOCI band reflectance in the morning time window and retains the instability of vegetation canopy spectra in the noon time window. The SANM provides a preliminary solution for normalizing the angles for the GSS RS data and makes the quantitative comparison of spatiotemporal RS data possible.

**Keywords:** angle normalization; vegetation canopy reflectance; geostationary satellite; path length correction; Minnaert model; GOCI

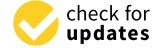

## 1. Introduction

A geostationary satellite (GSS) is characterized by a wide coverage area and strong maneuverability. It can realize minute-level high-frequency observations of specific areas, which greatly improves the efficiency of remote sensing (RS) data acquisition in cloudy and rainy areas [1]. Imaging sensors deployed on the traditional GSSs only have a single channel with a wide band range in the visible and near-infrared range (VNIR), and the spatial resolution is usually less than 1 km (e.g., the Fengyun-2 satellites [2] and the GOES (Geostationary Operational Environmental Satellite) generations before the GOES-R series). In recent years, imaging sensors deployed on GSSs have developed capabilities with multiple channels in the VNIR, and spatial resolutions have increased to 50–500 m (e.g., the COMS (Communication, Ocean, and Meteorological Satellite) [3], the Gaofen-4 satellite [4], the Fengyun-4 satellites [5], the Himawari-8 satellite [6], the GOES-R series,

the INSAT (Indian National Satellite System) satellite [7], the ELECTRO-L satellite [8], and the MTG (Meteosat Third Generation) satellite [9]). The optimized design of GSSs extends its application area from traditional meteorological, communications, and broadcasting to land-surface and ocean-water-color RS monitoring.

Vegetation is an important part of the Earth's ecosystem, and vegetation monitoring is the most complex part of land-surface RS monitoring. Vegetation has typical spectral characteristics and has a different canopy morphology due to differences in organizational structure, seasonal phase, and ecological conditions. Changes in canopy morphological features such as LAI (Leaf Area Index) and LAD (Leaf Angle Distribution) lead to changes in canopy porosity and extinction of cross-sectional size [10]. Therefore, it strongly influences the reflection and scattering characteristics in the optical and microwave bands, and this influence is perturbed by the terrain, illumination conditions, and observation geometry. Consequently, angle normalization should be urgently applied to RS monitoring of vegetation, which is better applied to monitoring land-surface phenology [11], biomass estimation [12], and surface vegetation patterns [13]. However, taking LAI and LAD as input parameters will reduce the usability of the angle normalization model: it is difficult to obtain ground observation of these features for large areas; remote sensing inversion products are obtained using remote sensing reflectance, and these products will introduce iteration errors. Therefore, it is necessary to use a simplified representation of BRDF (Bidirectional Reflectance Distribution Function).

The angle normalization of RS data, and of reflectivity in particular, consists of normalizing a uniform solar zenith angle and observation zenith angle, usually involving topographic correction (TC), solar angle correction or normalization (SAC), and detector angle correction or normalization (DAC). The digital elevation model (DEM)-based TC methods are the most widely used in the existing TC methods [14–16]. In recent years, many scholars have introduced non-Lambertian models and vegetation canopy structure parameters into TC methods to improve the accuracy of vegetation-canopy spectral topographic correction [17,18]. The existing SAC models use the cosine of the solar zenith angle as the main correction factor [19]. More complex algorithms introduced the intercept and slope for SAC models to solve the problem involving ground radiation signals in the presence of atmospheric scattering and refraction from the adjacent background, but no direct sunlight [20]. As for the DAC, only the 16-day synthetic products of MODIS (Moderate Resolution Imaging Spectroradiometer)/VIIRS (Visible Infrared Imaging Radiometer Suite) involving albedo and BRDF are currently widely recognized and applied. The spatial resolution of these products is 500 m, and the core of the production algorithm is the solution of a kernel-driven model [21,22].

The viewing angle on a per-pixel basis is constant, while the sun angle of GSS RS data changes from hour to hour, unlike those from sun-synchronous satellite sensors, and wide-field imaging characteristics magnify this difference [1], so it is urgent to normalize angles in the quantitative vegetation applications of GSS RS data. The operational BRDF and albedo algorithm uses a multi-day period of cloud-free angular surface reflectance that adequately samples the viewing geometry (at least seven observations) to fit an appropriate kernel-driven, RossThick-LiSparse-Reciprocal semi-empirical bidirectional reflectance model for the given surface location. However, the MODIS/VIIRS and sentinel-2A BRDF products have a lower spatial or temporal resolution, their applications are faced with the problem of spatial and temporal adaptability. Therefore, the research on angle normalization of RS data remains a hot topic and is the focus of this paper.

In this paper, the high-frequency and wide-field imaging characteristics of GSS sensors are fully exploited to propose a synthetic angle normalization model (SANM) for RS vegetation canopy reflectance. The GOCI (Geostationary, Ocean Color Imager) data obtained from GSS COMS were used to construct and verify the proposed SANM while considering the characteristics of diurnal variations in vegetation canopy spectra. The proposed SANM can provide a reference for the production of angle-normalization products for GSS RS data

and optimize the temporal resolution of angle-normalization products for RS vegetation canopy reflectance, which has important applications and practical significance.

## 2. Materials and Methods

### 2.1. Synthetic Angle Normalization Model Overview

The SANM proposed herein is based on the definition of angle normalization for RS data, using GSS RS data to get the normalized reflectance with the terrain slope, solar, and detector zenith angle are all 0°. The framework of the proposed model is presented schematically in Figure 1; the order of three core steps (TC, SAC, and DAC) was designed to satisfy the SAC and DAC models' assumption that the ground objects are aligned horizontally. Based on the literature research and comparison, the TC step uses the path-length correction (PLC) model, the SAC step uses the cosine of the solar zenith angle as the correction factor, and the DAC combines the imaging geometric coordinate rotation and the Minnaert model.

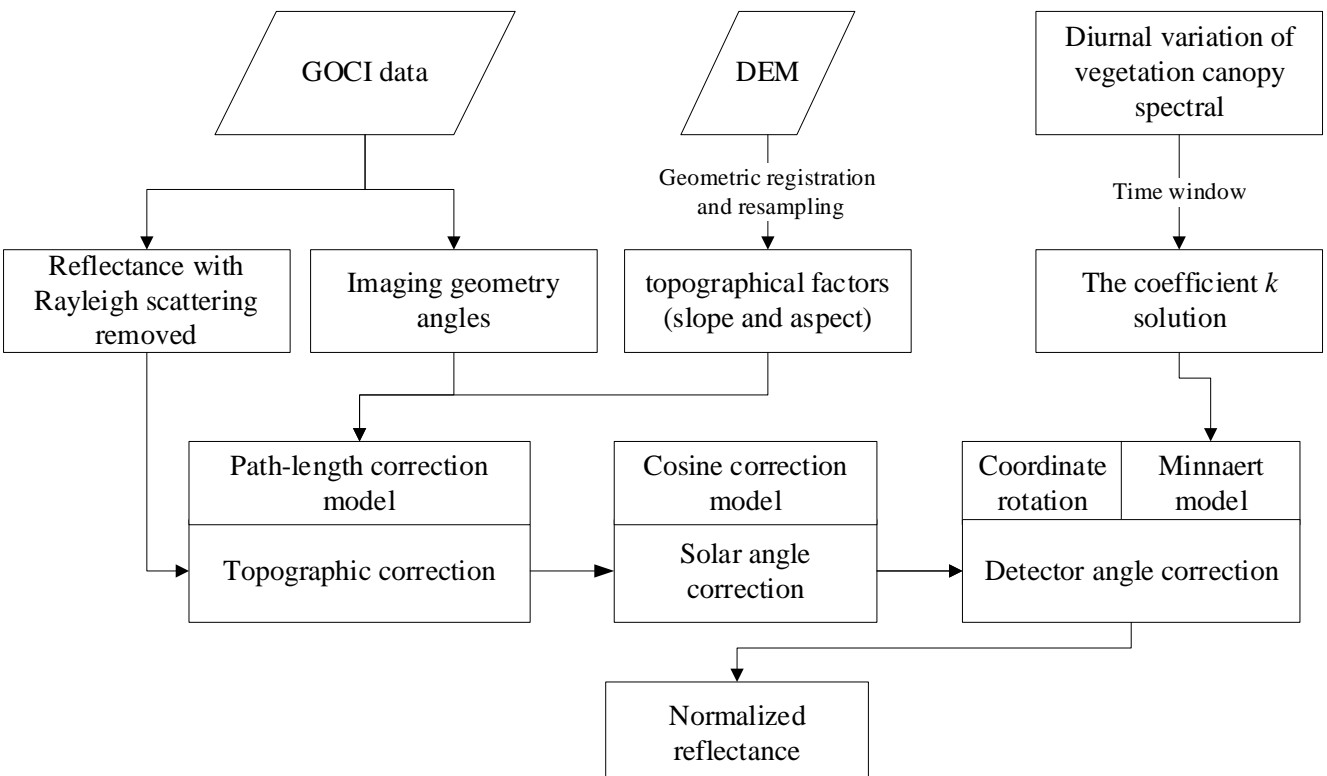

**Figure 1.** Schematic showing the workflow of the proposed method.

These three core steps are described in detail in the following three subsections. The angles and reflectance symbols used in each step and model application are described in Table 1.

The cosine of the angle between any two directions $\cos(\theta_{1-2})$ can be calculated as:

$$\cos(\theta_{1-2}) = \cos(\theta_1)\cos(\theta_2) + \sin(\theta_1)\sin(\theta_2)\cos(\varphi_1 - \varphi_2) \tag{1}$$

where $\theta_1$ and $\theta_2$ are zenith angles, and $\varphi_1$ and $\varphi_2$ are azimuth angles.

**Table 1.** Symbols used in the SANM.

| Symbol | Explanation |
| --- | --- |
| $\theta_S$ | Solar zenith angle |
| $\varphi_S$ | Solar azimuth angle |
| $\theta_D$ | Detector zenith angle |
| $\varphi_D$ | Detector azimuth angle |
| $\theta_T$ | Slope |
| $\varphi_T$ | Slope aspect |
| $\theta_{D-S}$ | Angle from observation direction to the solar incidence direction; derived from Equation (1) |
| $\theta_{S-T}$ | Angle from solar incidence direction to ground surface normal (solar incidence angle); derived from Equation (1) |
| $\theta_{D-T}$ | Angle from observation direction to ground surface normal; derived from Equation (1) |
| $\rho_t$ | Vegetation canopy reflectance observed by sensor |
| $\rho_{PLC}$ | Vegetation canopy reflectance after PLC model processing |
| $\rho_{pre}$ | Vegetation canopy reflectance after PLC model and SACM processing |
| $\rho_{Minnaert}$ | Vegetation canopy reflectance after Minnaert model processing |
| $\rho_{nom}$ | Vegetation canopy reflectance after SANM processing |

### 2.2. Topographic Correction for Vegetation Canopies-PLC

Vegetation grows geotropically; the terrain affects only the angle of the vegetation relative to the surface rather than the geometric relationship between the sun and the vegetation [23]. The TC method for vegetation canopies based on PLC [18] satisfies Assumption I, in which the radiance collected by the sensor is only from single scattering from leaves (i.e., the contributions from soil reflectance and from multiple scattering from leaves are negligible). In order to reduce the influence of mixed pixels and meet this assumption as far as possible, we select the mountainous area and field crop with full vegetation coverage to verify the algorithm. The relationship between $\rho_t$ and $\rho_{PLC}$ can be formulated as follows [18]:

$$\rho_{PLC} = \rho_t \frac{S_t(\varphi_S) + S_t(\varphi_D)}{S(\varphi_S) + S(\varphi_D)} \tag{2}$$

where $S(\varphi_S)$ and $S(\varphi_D)$ are the path lengths along the solar and viewing directions over flat terrain, respectively, and $S_t(\varphi_S)$ and $S_t(\varphi_D)$ are their counterparts over sloping terrain.

The path length along the direction of gravity is unity under any terrain conditions. The geometry of the extinction path at different angles is shown in Figure 2.

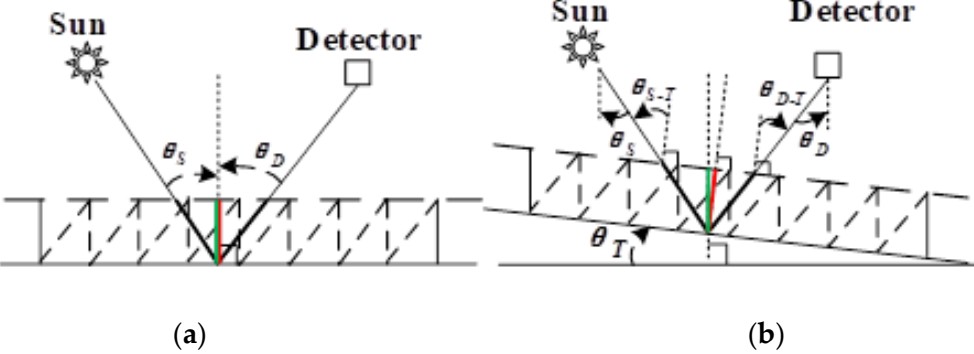

(**a**)        (**b**)

**Figure 2.** Path length of a (solar) beam through a canopy: (**a**) canopy on a horizontal surface; (**b**) canopy on an inclined surface. Green bold lines represent the path length along the zenith direction; it has unit magnitude. Red bold lines represent the path length along the direction normal to the vegetation canopy; its magnitude is $\cos(\theta_T)$. Black bold lines represent the path length (S) along an arbitrary direction in the vegetation canopy.

The path length in an arbitrary direction can be calculated as:

$$S(\theta_1, \varphi_1, \theta_T, \varphi_T) = \frac{\cos(\theta_T)}{\cos(\theta_{1-T})} = \frac{1}{\cos(\theta_1)[1 + \tan(\theta_1)\tan(\theta_T)\cos(\varphi_1 - \varphi_T)]} \tag{3}$$

where $\theta_1$ is $\theta_D$ or $\theta_S$, $\varphi_1$ is $\varphi_D$ or $\varphi_S$, and $\theta_{1-T}$ is $\theta_{S-T}$ or $\theta_{D-T}$.

### 2.3. Correction of Solar Angle

The solar angle includes the $\theta_S$ and the $\varphi_S$. The $\theta_S$ strongly influences the surface solar irradiance, whereas the $\varphi_S$ only affects the image detail [24]. Therefore, the existing SAC models only involves the $\theta_S$. Considering the BRDF characteristics of the land objects, we use the $\varphi_S$ to calculate $\theta_{D-S}$ as a comprehensive angle to carry out the alternative correction, see section "Correction of Detector Angle" for details.

The classical SAC model (SACM) formula is usually expressed as [25]:

$$\rho_{pre} = \rho_{PLC} / \cos(\theta_S) \tag{4}$$

### 2.4. Correction of Detector Angle

After the TC and SAC steps, $\rho_{pre}$ corrects for the influence of terrain and solar zenith angle, it does not take into account the difference in BRDF caused by imaging geometric differences between different phases. We rotated the coordinate to create an equivalent condition where the observation zenith angle is 0° (see Figure 3). Specifically, each pixel is simplified into a point object to ensure the BRDF character is unchanged; and finally, the four imaging geometric angles are converted to $\theta_{D-S}$ in DAC.

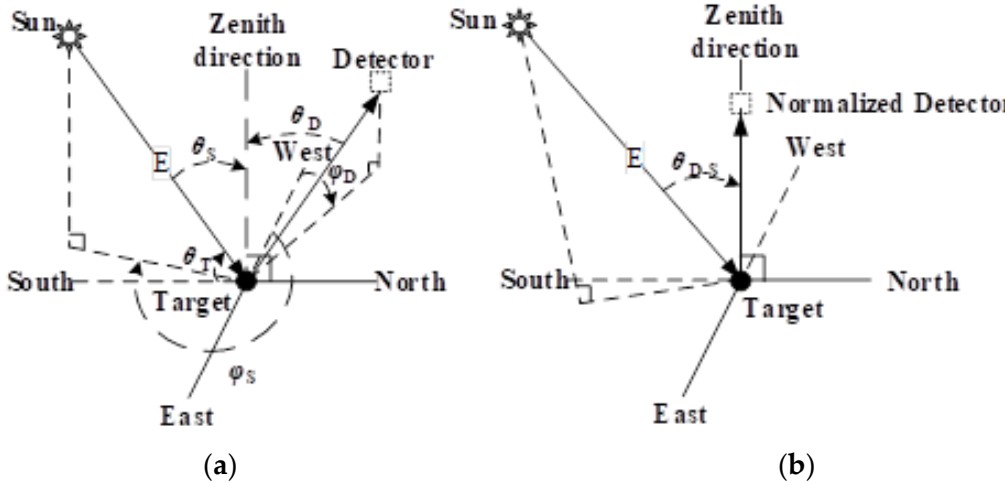

**Figure 3.** Schematic diagram of (**a**) the real imaging geometry and (**b**) the equivalent imaging geometry.

The Minnaert function was proposed for the TC of non-Lambertian albedo [26], where the k coefficient of the Minnaert function is the simplified representation of BRDF, and it is a constant in a given area. Note that the traditional k coefficient was found by simply applying a linear regression analysis with all types of objects in the Minnaert model early used for a single RS datum, and the optimized k coefficient was solved by applying a polynomial fit in the slope grading strategy in the modified Minnaert model to better represent the terrain change. However, the above Minnaert model ignores the influence of the ground object on the k coefficient. In this paper, the k coefficient is solved pixel by pixel using the high-frequency imaging feature of GSS RS data.

The DAC formula based on coordinate rotation and the Minnaert model can be expressed as:

$$\rho_{Minnaert} = \rho_{pre} \cos(\theta_T) / [\cos(\theta_T)\cos(\theta_{D-S})]^k \tag{5}$$

The $\theta_T$ of each pixel has been corrected to $0°$ after the application of Equation (4), so Equation (5) can be further reduced to:

$$\rho_{nom} = \rho_{pre} / [\cos(\theta_{D-S})]^k \qquad (6)$$

Note that the diurnal variation in the vegetation canopy spectra based on field experiments [27] and related studies [28] shows that the local time period before 11:00 (called the morning time window) and after 13:30 (called the afternoon time window) are the periods when the vegetation canopy spectrum itself is relatively stable; whereas the local time period from 11:00 to 13:30 (called the noon time window) is when the vegetation canopy spectrum changes drastically. Thus, to ensure that the vegetation canopy spectrum itself is relatively stable for data screening, we find the k coefficient as a function of the time window for each pixel.

### 2.5. Study Area and Data

The study area was located at the junction point of Jiangsu province and Anhui province of China (117°57′43″ E~118°38′13″ E, 32°09′43″ N~32°24′14″ N) (see Figure 4a). The study area spans in altitude from −52 to 392 m (see Figure 4b), and its slope ranges from 0° to 30°. The conventional crops include wheat, rice, rapeseed, soybean, etc., and forests include poplar, Masson pine, etc.

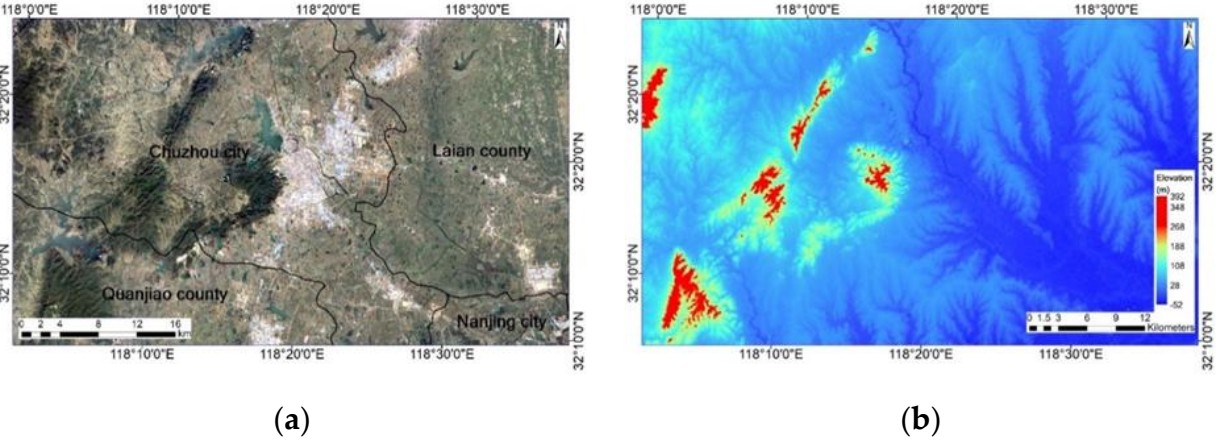

**Figure 4.** (**a**) Geographic location and (**b**) DEM of the study area.

The GOCIs onboard the Communication, Ocean, and Meteorological Satellite (COMS), observation area of 2500 × 2500 km is centered on the Korean Peninsula (130° E, 36° N) and supports a spatial resolution of 500 m; the spectral features are shown in Table 2. The GOCI is capable of producing images at hourly intervals and receives images eight times a day from 08:15 to 15:45 CST (China Standard Time UT + 8:00).

**Table 2.** GOCI satellite band parameter information.

|                  | B1  | B2  | B3  | B4  | B5  | B6  | B7  | B8  |
|------------------|-----|-----|-----|-----|-----|-----|-----|-----|
| Band length (nm) | 412 | 443 | 488 | 555 | 660 | 680 | 745 | 865 |
| Band width (nm)  | 20  | 20  | 20  | 20  | 20  | 10  | 20  | 40  |

The GOCI images acquired on 22 April 2015 were used for model verification because of the advantageous winter wheat growth cycle and the good spatial distribution of the cloud coverage for the GOCI images. In the study area, 22 April 2015 was during the jointing stage of winter wheat; the crops appeared to be growing well with full ground coverage. However, the GOCI images received after 14:00 CST on 22 April 2015 suffered from thin cloud coverage in the study area, so these two images were not used in the data

processing and analysis. Subsequently, each image is represented by the imaging hour in CST.

We first used the GDPS (the GOCI data-processing system) to process GOCI L1B data to obtain the Rayleigh-corrected reflectance, the latitudes and longitudes of the four corner points and the center point, the solar angles and observation angles of each pixel, etc. We then subset the reflectance products according to the coordinate range of the study area. Furthermore, for comprehensive considerations of the synchronous ground observation experiment on winter wheat [29] and the GOCI pixel NDVI covering the samples, we took 0.6 as the NDVI threshold and used LAND_NDVI products to screen the ground object type, and the model was applied only to the 5292 selected vegetation pixels. Finally, after projection conversion and resembling operations, the 90 m Chinese resolution digital elevation data product was used to calculate the topographical factors (slope and aspect) of each pixel of the GOCI reflectance products after geometric registration and resampling.

### 2.6. Method Evaluation Strategies

Numerous strategies have been used to assess the performance of topographic correction methods and solar normalization methods [18,30]. To obtain an objective evaluation, we used three different methods:

(i)   Correlation analysis between reflectance in different imaging periods. Because the vegetation canopy spectrum is relatively stable in the morning time window, the effective angle normalization model should strengthen the reflectance correlation of different imaging phases in the morning time window and make the slope of the linear regression equation closer to unity. Conversely, the vegetation canopy spectrum changes drastically in the noon time window, so the effective angle normalization model should weaken the reflectance correlation of different imaging phases and make the slope of the linear regression equation further depart from unity.

(ii)  Analysis of the correlation between the cosine of the imaging geometry angles and reflectance. This is one of the most widely used quantitative evaluation methods. The efficiency of the normalization methods can be quantified by using $R^2$ and the imaging geometry angles of the corresponding linear regression. The ideal normalization method should make $R^2$ approach zero [31].

(iii) Radiometric stability. Theoretically, the maximum (minimum) reflectance in the original image before correction should appear in the sunny (shady) slope and will decrease (increase) after topographic correction. Consequently, a successful correction method will reduce the reflectance range. Moreover, the median reflectance is relatively stable and invariable after correction [30].

## 3. Results

According to the typical vegetation spectral characteristics, the bands 400–730 nm and 730–900 nm are two typical spectral bands in the winter wheat canopy spectrum [29]. GOCI band 5 (650–670 nm) and band 8 (845–885 nm) are used to produce NDVI (Normalized Difference Vegetation Index) products and were selected for model application analysis.

### 3.1. Correlation between Different Imaging Phases

To comprehensively compare how normalizing the angles affects the treatment of the models of the GOCI reflectance bands, Table 3 shows the detailed regression results for the band 5 reflectance and band 8 reflectance for different imaging hours.

Table 3 shows that the correlations for the band 8 reflectance between different imaging phases are significantly better than for the band 5 reflectance in the corresponding phases, which is consistent with the diurnal variation in the field-measured reflectance spectra of the vegetation canopy [29]. The slope of the linear fit and $R^2$ in Table 3 further indicates that the normalization has no effect on the results of the PLC model for the GOCI reflectance correlation between different imaging phases, the SACM suffers from over-correction, and

the SANM not only significantly reduces the over-correction of the SACM but also preserves the instability of the vegetation canopy reflectance spectra in the noon time window.

**Table 3.** Slope and $R^2$ of fit for GOCI band 5 reflectance and GOCI band 8 reflectance between different imaging times.

| | Imaging Hour | Linear Fit | Band 5 Reflectance | | | | Band 8 Reflectance | | | |
|---|---|---|---|---|---|---|---|---|---|---|
| | | | Ori | PLC | SACM | SANM | Ori | PLC | SACM | SANM |
| Morning window | 08–09 | Slope | 1.014 | 0.998 | 0.802 | 0.976 | 0.924 | 0.917 | 0.731 | 1.021 |
| | | $R^2$ | 0.768 | 0.767 | 0.771 | 0.889 | 0.901 | 0.900 | 0.901 | 0.951 |
| | 08–10 | Slope | 1.144 | 1.116 | 0.785 | 1.008 | 0.857 | 0.848 | 0.589 | 0.994 |
| | | $R^2$ | 0.844 | 0.832 | 0.847 | 0.993 | 0.862 | 0.856 | 0.860 | 0.997 |
| | 09–10 | Slope | 0.958 | 0.957 | 0.831 | 0.891 | 0.904 | 0.903 | 0.785 | 0.914 |
| | | $R^2$ | 0.792 | 0.794 | 0.793 | 0.833 | 0.906 | 0.906 | 0.906 | 0.925 |
| Noon window | 11–12 | Slope | 0.804 | 0.808 | 0.797 | 0.786 | 0.877 | 0.878 | 0.871 | 0.814 |
| | | $R^2$ | 0.705 | 0.708 | 0.705 | 0.718 | 0.871 | 0.872 | 0.871 | 0.855 |
| | 11–13 | Slope | 0.847 | 0.849 | 0.887 | 0.801 | 0.859 | 0.859 | 0.902 | 0.689 |
| | | $R^2$ | 0.324 | 0.327 | 0.328 | 0.368 | 0.798 | 0.799 | 0.798 | 0.703 |
| | 12–13 | Slope | 0.848 | 0.849 | 0.890 | 0.846 | 0.962 | 0.962 | 1.016 | 0.860 |
| | | $R^2$ | 0.298 | 0.302 | 0.298 | 0.353 | 0.883 | 0.884 | 0.884 | 0.850 |

*3.2. Sensitivity to Imaging Geometry Angles*

Band 8 normalization has a consistent effect with band 5, but with higher reflectance, so we take band 5 reflectance from 08:15 CST as an example; Figure 5 compares the cosine of the imaging geometry angle with the reflectance before and after each normalization model (i.e., the PLC model, the SACM, and the proposed SANM).

The correlation is extremely weak between the original band 5 reflectance with $\cos(\theta_{S-T})$, $\cos(\theta_{D-T})$, and $\cos(\text{slope})$: $R^2$ for the linear fit is $2.88 \times 10^{-4}$ (see Figure 5a), 0.001 (see Figure 5e), and 0.011 (see Figure 5i). These results are attributed to the small difference in imaging geometry when the study area is small. The use of the PLC model significantly improves the correlation between the band 5 reflectance and $\cos(\theta_{S-T})$ and $\cos(\theta_{D-T})$: $R^2$ for the linear fit increased to 0.038 (see Figure 5b) and 0.03 (see Figure 5f). The use of the SACM significantly reduced the correlation between band 5 reflectance and $\cos(\theta_{S-T})$ and $\cos(\text{slope})$: $R^2$ for the linear fit decreased to $1.402 \times 10^{-4}$ (see Figure 5c) and 0.01 (see Figure 5k). The use of the SANM significantly improved the correlation between band 5 reflectance and $\cos(\theta_{D-T})$: $R^2$ for the linear fit increased to 0.015 (see Figure 5h) from the original 0.001 (see Figure 5e); whereas the correlation is significantly reduced between band 5 reflectance and $\cos(\theta_{S-T})$ and $\cos(\text{slope})$: $R^2$ for the linear fit decreased to $1.6 \times 10^{-4}$ (see Figure 5c) and 0.004 (see Figure 5k). These results indicate that the normalization by SANM proposed herein has a better effect on the solar angle of incidence and slope (i.e., a lower $R^2$); however, it presents a poor normalization effect on $\theta_{D-T}$.

*3.3. Radiometric Stability*

Theoretically, after correction, the reflectance ranges should be contained in their counterparts before correction [30]. Box plots of band 5 reflectance and band 8 reflectance from the uncorrected and corrected images shows that each angle normalization for a given model has the same effect on the reflectance of bands 5 and 8, and the reflectance distribution is more concentrated when the mean reflectance is lower (see Figure 6). Figure 6 also shows that the PLC model did not change the distribution of the GOCI band reflectance and the variations in the imaging phases: the SACM suffered from over-correction, which increased with the solar zenith angle, and the SANM significantly improved the over-correction problem of the SACM. The band reflectances were stable in the morning time window and decreased in the noon time window after SANM processing, which is consistent with the intraday variation of the field-measured reflectance spectra of the vegetation canopy [28].

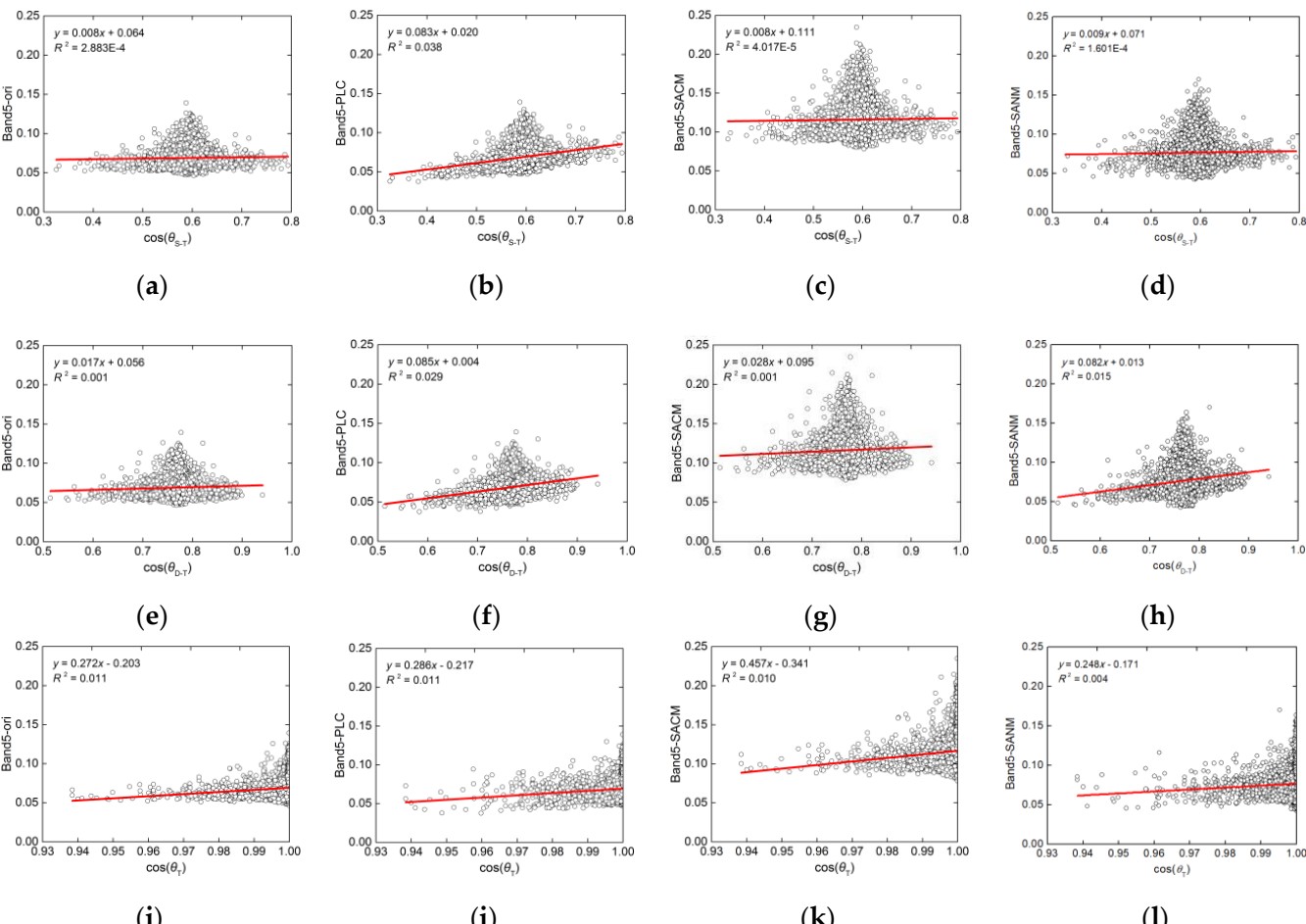

**Figure 5.** (**a**) Density scatter plots between the original Rayleigh-corrected GOCI band 5 reflectance (Band5-ori) at 08:15 CST and $\cos(\theta_{S-T})$, the red line is linearly fit to data; (**b**) Same as (**a**) except using the GOCI band 5 reflectance after PLC correction (Band5-PLC); (**c**) Same as (**a**) except between the GOCI band 5 reflectance after ASCM correction with Band5-ori as $\rho_t$ (Band5-SACM); (**d**) Same as (**a**) except between the GOCI band 5 reflectance after SANM normalization (Band5-SANM); (**e**) Same as (**a**) except using the $\cos(\theta_{D-T})$; (**f**–**h**) Same as (**e**) except using Band5-PLC, Band5-SACM, and Band5-SANM, respectively; (**i**) Same as (**a**) except using the $\cos(\theta_T)$; (**j**–**l**) Same as (**i**) except using Band5-PLC, Band5-SACM, and Band5-SANM, respectively.

Figure 6a shows that the mean original reflectance of band 5 increases from 08:15 to 11:15 CST, after which it decreases. After PLC processing, band 5 reflectance underwent no significant change in range or distribution (Figure 6b) compared with Figure 6a and retained the variations of band 5 reflectance for the various imaging phases. Figure 6c shows that, because of the over-correction problem, large differences exist in the band 5 reflectance after SACM processing. The mean reflectance of band 5 after SACM processing decreased along the imaging phase and increased up to the 04 phase. Figure 6d shows that the band 5 reflectance in the 00 phase to the 03 phase are more similar after SANM processing, and the band 5 reflectance in the 04 phase and 05 phase are lower than those for the other phases after SANM processing. Since all selected pixels found almost no shadows, the reduction in the reflectance range is inconspicuous.

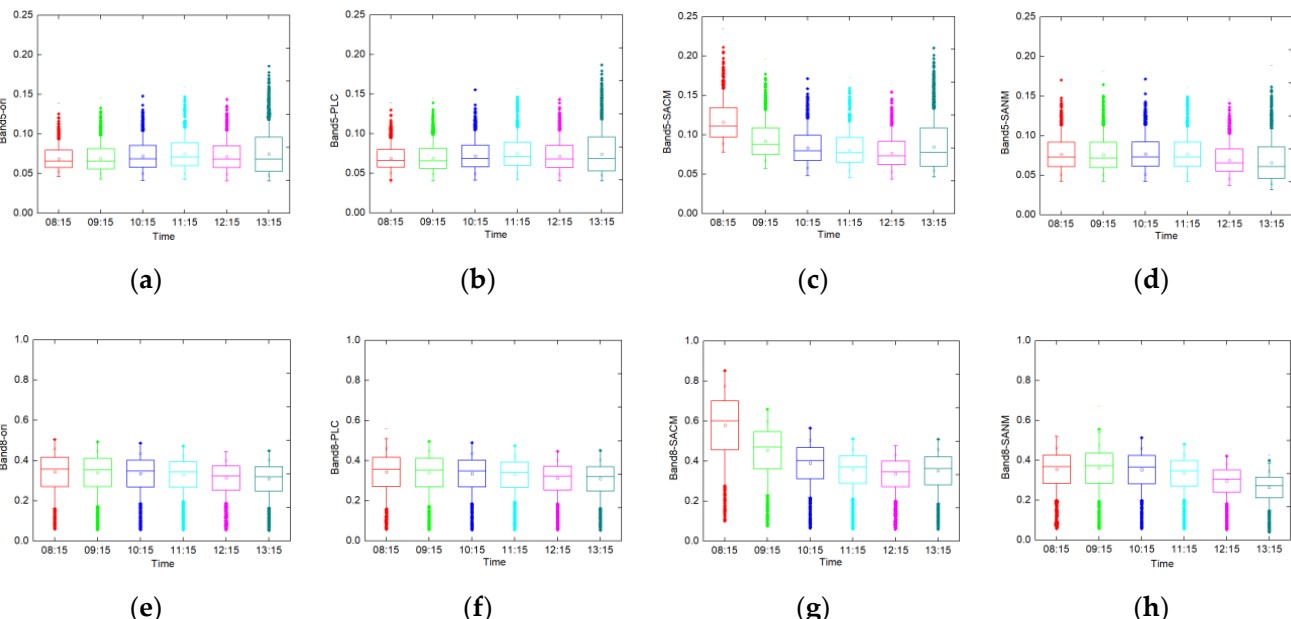

**Figure 6.** Box plots of (**a**) the original Rayleigh-corrected GOCI band 5 reflectance; (**b**) the GOCI band 5 reflectance after PLC correction (Band5-PLC); (**c**) the GOCI band 5 reflectance after ASCM correction with Band5-ori as $\rho_t$ (Band5-SACM); (**d**) the GOCI band 5 reflectance after SANM normalization (Band5-SANM) in each imaging time for the morning time window and the double-peak time window; (**e**–**h**) same as (**a**–**d**) correspondingly except using band 8 reflectance.

## 4. Discussion

With the help of GSS RS data, the proposed SANM can improve the time resolution of angle-normalized products for RS reflectance from a vegetation canopy to the hourly level. However, the following problems exist in the model-construction process:

In the TC step, the extinction path-length formula is derived as a hypothetical condition for a dense canopy without considering the effects of a sparse canopy [32]. However, the actual vegetation canopy structure usually has daily, quarterly, and annual variations and regional differences, which strongly impact the BRDF and biomass retrieval [33]. In the subsequent model optimization, we propose to introduce vegetation cover factor variables to distinguish how a dense canopy versus a sparse canopy affects the reflectance spectrum from a vegetation canopy [34,35].

In the SAC step, we used the simplest cosine correction model and did not consider whether the fit to the reflectance and cosine of the solar zenith angle passes through the origin, which depends on atmospheric scattering and refraction from adjacent pixels [36]. Subsequent research should introduce the intercept and slope into the SAC step for optimization. However, the difficulty is the determination of the intercept, especially in the case of large changes in solar angle caused by the wide field and high frequency of GSSs.

In the DAC step, we set the time window when solving for the Minnaert model k coefficient. The given time window only considered few a diurnal variations of the reflectance spectra from the vegetation canopy, which limited the effective data used for calculating the k coefficient. In the follow-up study, a database will be created of the diurnal variations of reflectance spectra from canopies of different types of vegetation through literature research and field measurements so as to screen data to more accurately solve for the k coefficient. In addition, the distribution of the k coefficient obtained herein exceeded the conventional range of 0–1 (see Figure 7), which may be due to the GSS imaging regions that are located in the backscattering area [26]. Subsequent research should study how the scattering orientation affects the k coefficient and determine the range of the k coefficient under the conditions of fixed observation angle.

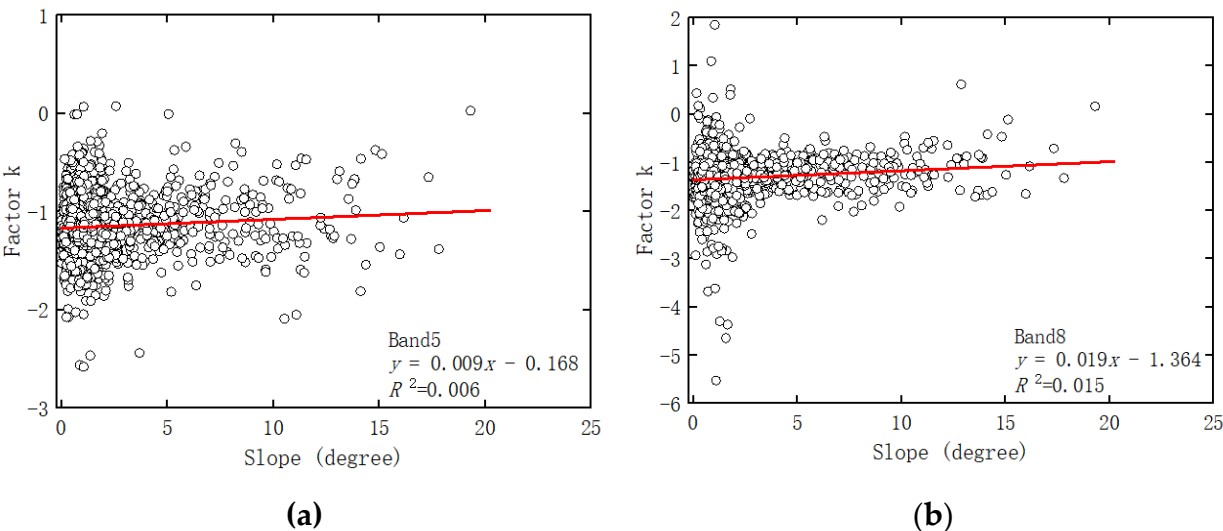

**Figure 7.** Density scatterplots between the slope and factor k in SANM for GOCI images band 5 (**a**) and band 8 (**b**) ground object reflectance.

## 5. Conclusions

By using the PLC model, the cosine model for solar-angle normalization, and the Minnaert model, we herein establish a SANM for the reflectance of the GSS RS vegetation canopy. GOCI images were used to test the SANM, and a multi-criteria analysis was used in the evaluation. With the PLC model, normalization has no effect on the correlation of the GOCI reflectance between different imaging phases. However, the correlation is significantly improved between the band reflectance with the cosine of the solar angle of incidence and the cosine of the angle from the observation direction to the ground surface normal. The SACM significantly reduced the correlation between band 5 reflectance with the cosine of the solar angle of incidence and slope, but it suffered from over-correction. Whereas the SANM significantly improved the over-correction problem for the SACM and also preserved the instability of the vegetation canopy spectra in the noon time window. The use of the SANM significantly reduced the correlation between the band reflectance with the cosine of the solar angle of incidence and the slope. For normalizing the angle of the high-frequency GSS RS, the SANM outperformed all other methods, which indicates that it has a strong potential for applications and for monitoring land-surface phenology, estimating biomass, etc.

**Author Contributions:** Conceptualization, Y.L. (Yinghao Lin) and Q.T.; methodology, Y.L. (Yinghao Lin) and B.Q.; software, Y.W. and Y.X.; validation, Y.L. (Yinghao Lin), X.Z. and Y.W.; writing—original draft preparation, Y.L. (Yinghao Lin) and Q.T.; writing—review and editing, Y.L. (Yinghao Lin) and Y.L. (Yang Lian); visualization, Y.L. (Yinghao Lin) and Y.X.; funding acquisition, Y.L. (Yinghao Lin) and Y.W. All authors have read and agreed to the published version of the manuscript.

**Funding:** This research was funded by Key R&D and Promotion Projects of Henan Province, grant number 222102320163; China High-resolution Earth Observation System, grant number 80-Y50G19-9001-22/23; Major Project of Science and Technology of Henan Province, grant number 201400210300; National Natural Science Foundation of China, grant number 42071318; National Defense Basic Research Projects of China, grant number JCKY2020908B001; National Basic Research Program of China, grant number 2019YFE0126600 and Kaifeng science and technology development plan, grant number 2002001.

**Institutional Review Board Statement:** Not applicable.

**Informed Consent Statement:** Not applicable.

**Data Availability Statement:** The GOCI data and GDPS can be downloaded from: http://kosc.kiost.ac.kr/index.nm?menuCd=54&lang=en (accessed on 10 June 2022).

**Acknowledgments:** The author would like to thank all contributors to this study.

**Conflicts of Interest:** The authors declare no conflict of interest.

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
