# Peer review of "A Synthetic Angle Normalization Model of Vegetation Canopy Reflectance for Geostationary Satellite Remote Sensing Data"

_agriculture, doi:10.3390/agriculture12101658_

Round 1

Reviewer 1 Report

The paper proposed a new method for radiometric correction of the geostationary image. The manuscript is well-written; nevertheless, I have some concerns about the proposed method and its implementation, which require more clarification.  

Although not explicitly mentioned, one of the assumptions of the proposed algorithm is assuming a full coverage of plant canopies. there are mixed pixels for crops, particularly in this resolution. Hence, the authors should design an experiment to show how this issue affects the algorithm.

For crops farm, the land is usually leveled before plantation. In this case, why do we need topography correction using PLC?

On page 7, the authors mentioned that the ideal method has zero R2. The R-squared or goodness of fit is better when it is closest to one; unless another definition is used. Please double-check this issue.

The main goal of all radiometric corrections is to obtain surface reflectance. Based on the authors' claim, surface reflectance was available. Why it has not been used for accuracy assessment?

In the discussion section, the authors listed some shortcomings of the method, which is very informative. But why they are not addressed in the method? In addition, the last paragraph is more similar to a reviewer’s comment than a discussion.

Author Response

Response to Reviewer 1 Comments

The paper proposed a new method for radiometric correction of the geostationary image. The manuscript is well-written; nevertheless, I have some concerns about the proposed method and its implementation, which require more clarification.

Although not explicitly mentioned, one of the assumptions of the proposed algorithm is assuming a full coverage of plant canopies. there are mixed pixels for crops, particularly in this resolution. Hence, the authors should design an experiment to show how this issue affects the algorithm.

Response 1: We conducted a synchronous ground observation experiment on winter wheat (the crops appeared to be growing well with full ground coverage) [1], and determined the NDVI threshold of 0.6 for vegetation pixels selection combining the ground observation NDVIs and GOCI pixel NDVI covering the samples, thus, the coverage of the pixel is guaranteed. This explanation is supplemented in section 2.5 of the paper.

For crops farm, the land is usually leveled before plantation. In this case, why do we need topography correction using PLC?

Response 2: On the one hand, geostationary satellite can not realize the zenith observation, but keeps the large inclination angle observation. On the other hand, solar angle varies at different phases in the same area. These two aspects will lead to the difference of imaging geometry in different regions and phases, therefore even the leveled farm also need topography correction using PLC, this can improve the quantitative comparability for the GSS RS products in different regions.

On page 7, the authors mentioned that the ideal method has zero R2. The R-squared or goodness of fit is better when it is closest to one; unless another definition is used. Please double-check this issue.

Response 3: This two evaluation criteria are not for the same result data, ‘the ideal method has zero R2’ is coresponding to the second strategy in 2.6 (Analysis of correlation between the cosine of the imaging geometry angles and reflectance), and ’ The R-squared or goodness of fit is better when it is closest to one’ is coresponding to the first strategy in 2.6 (Correlation analysis between reflectance in different imaging periods).

The main goal of all radiometric corrections is to obtain surface reflectance. Based on the authors' claim, surface reflectance was available. Why it has not been used for accuracy assessment?

Response 4: The ground observation experiment was carried out only on a small part of the Wangbodang farm, mainly to analysis the diurnal change characteristics of canopy spectral reflectance and vegetation indices of winter wheat in the jointing stage, this cannot satisfy the ground validation of large area angle normalization. This paper focuses on verifying whether the proposed model can effectively weaken the reflectance correlation of different imaging phases and correlation between the cosine of the imaging geometry angles and reflectance, in order to preprocess for multi-temporal remote sensing data collaborative application.

In the discussion section, the authors listed some shortcomings of the method, which is very informative. But why they are not addressed in the method? In addition, the last paragraph is more similar to a reviewer’s comment than a discussion.

Response 5: Model optimizations in the discussion section need additional ground observation experiments which is under going. At present ,we are conducting and publishing papers on these research in turn, outcomes involve optimal Minnaert topographic correction model based on land cover classification, BRDF prototype library construction and so on. The lass paragraph will be deleted in the paper.

References

  1. Lin, Y.; Shen, H.; Tian, Q.; Gu, X.; Yang, R.; Qiao, B. Mechanisms underlying diurnal variations in the canopy spectral reflectance of winter wheat in the jointing stage. Sci. 2020, 118, 1401–1406, doi:10.18520/cs/v118/i9/1401-1406.

Reviewer 2 Report

The method implemented happens enough effective and operative to be advantageously used to significantly improve the exploitation of visible and NIR reflectance data derived from GSS images for suitable assessment of vegetation biophysical parameters, but in my opinion some aspects about the methodology and quantification of improvement obtained should be better explained and supported.  In particular the following points should be enhanced and deepened:

·         the distribution results of variable Minnaert k assessment in terms of regression parameters distribution should be reported explained and discussed;

·         comprehensive data referring to second and third method for the quantitative assessment of the improvements obtained, should be reported and discussed:

o    in addition to those of band 5 (Fig. 5), also data related to band 8 normalization should be included;

to better highlight the reflectance range reduction according to the third method, in addition to graphs boxplot of figure a tabular data should be reported and discussed.  

Author Response

Response to Reviewer 2 Comments

The method implemented happens enough effective and operative to be advantageously used to significantly improve the exploitation of visible and NIR reflectance data derived from GSS images for suitable assessment of vegetation biophysical parameters, but in my opinion some aspects about the methodology and quantification of improvement obtained should be better explained and supported.  In particular the following points should be enhanced and deepened:

the distribution results of variable Minnaert k assessment in terms of regression parameters distribution should be reported explained and discussed;

Response 0-1: The distribution results of variable Minnaert k is supplemented in the last paragraph of section 4 of the paper.

comprehensive data referring to second and third method for the quantitative assessment of the improvements obtained, should be reported and discussed:

Response 0-2: Comprehensive data referring to second and third method for the quantitative assessment coresponding to section 3.2 and 3.3 respectively.

in addition to those of band 5 (Fig. 5), also data related to band 8 normalization should be included;

Response 0-3: Band 8 normalization has consistent effect with band 5 but with higher reflectance, so band 8 normalization was lacking considering space. This explanation is supplemented in section 3.2 of the paper.

to better highlight the reflectance range reduction according to the third method, in addition to graphs boxplot of figure a tabular data should be reported and discussed. 

Response 0-4: Boxplot graphs of different bands used uniform unit of length separately, this can achieve similar effect in highlight the reflectance range reduction, and no duplicate information in the paper.

Finally, according to the conclusion reported, the effects of a preventive classification introduced in the proposed methodology to distinguish the different vegetation cover in order to better support the proper assessment of k coefficient should be briefly discussed.

Response 0-5: Outcomes of optimal Minnaert topographic correction model based on land cover classification have been accepted by National Remote Sensing Bulletin. Considering that the coefficient k of the Minnaert function is the simplified representation of BRDF, these outcomes proposes an optimal Minnaert topographic correction model based on land cover classification, referred to as CMinnaert model. In the pre classification of land cover types, two classification schemes are applied: classification based on the first level standard of 《Current land use classification》 and classification based on vegetation density, so as to verify the stability of CMinnaert model and identify the preferred classification scheme of land cover types. Results show that it choose classification based on the first level standard for pre classification when using CMinnaert model. In this paper, the coefficient k is found for each pixel by using the high-frequency imaging feature of GSS RS data, so pre-classification is eliminated. Related explanation is in the paragraph under Figure 3.

Detailed comments follow.

  1. Pag 2, lines 55-56, 1. Introduction, These features (LAI/LAD) characterize the different vegetation types and greatly affect their BRDF. Why You didn't consider this aspect in your angle normalization methodology? An ampler explanation and justification must be provided;

Response 1: Control of the input parameters is the original intention of the model, we used the coefficient k of the Minnaert function to simplified represent BRDF. The most essential, these features will reduce the usability of the model: it is difficult to obtain ground observation of these features in large area; remote sensing inversion products are obtained used remote sensing reflectance, these products will introduce iteration errors. This explation is supplemented in the second paragraph of section 1 of the paper.

  1. Pag 4, lines 122-126, 2. Materials and Methods, 2.2. Topographic Correction for Vegetation Canopies-PLC. The single scattering assumption for NIR reflectance from high coverage vegetation canopies is weak. In addition, the number of pixel selected satisfying this condition (fully vegetated) should be reported;

Response 2: We conducted a synchronous ground observation experiment on winter wheat (the crops appeared to be growing well with full ground coverage) [1], and determined the NDVI threshold of 0.6 for vegetation pixels selection combining the ground observation NDVIs and GOCI pixel NDVI covering the samples, thus, the coverage of the pixel is guaranteed. This explanation is supplemented in section 2.5 of the paper.

  1. Pag 4, lines 162-175, 2. Materials and Methods, 2.4. Correction of Detector Angle. Is the k coefficient dependent also on the specific vegetation land cover? The assessment methodology of k coefficient from multi temporal GSS RS data must be better explained and described including the results obtained (i.e. regression R2 and p-value parameters mean values), taking into account the morning time window selected. Possibly the BRDF correction formulation of the GSS data should by provided as BRDF multiplicative coefficient of reflectance detected by the sensor in order to be compatible with the expression available for polar sensors (i.e. MODIS);

Response 3: GSS data have enough observation angle within a day and can realize the the coefficient k solution pixel by pixel. We conducted a synchronous ground observation experiment on winter wheat to analysis the diurnal change characteristics of canopy spectral reflectance and vegetation indices of winter wheat in the jointing stage, the GOCI data are divided into time windows according to this diurnal change characteristics. Polar sensor data acquisition period is in days, it should consider time window according to phenological regularity.

  1. Pag 6, lines 178-182, 2. Materials and Methods, 2.5. Study Area and Data. The study area is about 50x50 Kmq and 100x100 image pixels. The BRDF reflectance anisotropy varies depending on the specific land cover and density , in particular kind of vegetation (with its specific LAI / LAD parameters ) or other ( i.e. urban visible in the image reported ). Please report the number of pixels referring to fully vegetated areas You selected for Your normalization methodology implementation;

Response 4: We conducted a synchronous ground observation experiment on winter wheat (the crops appeared to be growing well with full ground coverage) [1], and determined the NDVI threshold of 0.6 for vegetation pixels selection combining the ground observation NDVIs and GOCI pixel NDVI covering the samples, thus, the coverage of the pixel is guaranteed. This explanation is supplemented in section 2.5 of the paper.

  1. Pag 6, lines 183-188, 2. Materials and Methods, 2.5. Study Area and Data. Please include the spectral features (i.e. acquisition bands) of GOCI sensor and its orbital characteristics;

Response 5: The orbital characteristics ‘The GOCI, onboard the Communication, Ocean and Meteorological Satellite (COMS), observation area of 2500 km × 2500 km is centered on the Korean Peninsula (130°E, 36°N)’ is in the paragraph following Figure 4. The spectral features (i.e. acquisition bands) of GOCI sensor and its orbital characteristics is supplemented as the new Table 2.

  1. Pag 6, lines 201-205, 2. Materials and Methods, 2.5. Study Area and Data. How did You select vegetation pixels through NDVI ? Which are the threshold values? How many pixels of the GOCI image shown ( fig 4 about 100X100 pixels) did You analyse? The registration and resampling of the GSS data to be compatible with DEM should be better explained and results in terms of RMSE must be reported;

Response 6: We conducted a synchronous ground observation experiment on winter wheat (the crops appeared to be growing well with full ground coverage) [1], and determined the NDVI threshold of 0.6 for vegetation pixels selection combining the ground observation NDVIs and GOCI pixel NDVI covering the samples, thus, the coverage of the pixel is guaranteed. This explanation is supplemented in section 2.5 of the paper.

  1. Pag 7, lines 242-250, Table 2. 3. Results, 3.1. Correlation between different imaging phases. The pourer results obtained for the images of the noon window should be better explained and justified;

Response 7: We conducted a synchronous ground observation experiment on winter wheat, results showed that, the canopy spectral in the noon window showed double-peak trend, this may caused by midday depression of photosynthesis [1]. This explanation is supplemented in the paragraph following the new Talbe 3 of the paper.

  1. Pag 8, lines 254-257, Figure 5. 3. Results, 3.2. Sensitivity to imaging geometry angles. In addition to those of band 5 in the visible range, the results obtained for the GSS reflectance data in Band 8 (NIR) should be reported and discussed also;

Response 8: Band 8 normalization has consistent effect with band 5 but with higher reflectance, so band 8 normalization was lacking considering space. This explanation is supplemented in section 3.2 of the paper.

  1. Pag 8, lines 279-280, Figure 5. 3. Results, 3.2. Sensitivity to imaging geometry angles. What do You mean for: “ ….the normalization of should be optimized”. A better explanation should be included;

Response 9: The use of the SANM significantly improved the correlation between band 5 reflectance and cos():  for the linear fit increased to 0.015 (see Figure 5h) from the original 0.001 (see Figure 5e). The ideal normalization method should make  approach zero, so the proposed model present poor normalization effect on , which need optimization. This explanation is modified in the paragraph following Figure 5 of the paper.

  1. Pag 9, lines 308-310, Figure 6. 3. Results, 3.3. Radiometric stability. Why You did not report the results related to 3th methods to verify the improvement of the normalized GSS data? In particular, the reduction of the reflectance range of corrected data should be shown. Why this reduction doesn’t happen in the boxplot graphs in figure 6? An explanation and justification of this aspect should be provided;

Response 10: We checked the correction pixels and found almost no shadows, so this reduction doesn’t happen in the boxplot graphs in figure 6. This explanation is supplemented in section 3.3 of the paper. Recently, we focus on the research of Optimized Topographical Correction Method Combined Minnaert Model with Shadow Factor using Landsat 8/OLI data, we selection similar coverage pixels in sunny, shadow and falling shadow to test the new model. Results showed that Landsat 8/OLI data exist obvious shadow and falling shadow area, and difference of NDVIs of selected pixels significantly reduced.

  1. Pag 9, lines 316-318,4. Discussion. Which NDVI threshold provides the selection of a suitable number of vegetated areas with enough dense canopies?

Response 11: We conducted a synchronous ground observation experiment on winter wheat (the crops appeared to be growing well with full ground coverage) [1], and determined the NDVI threshold of 0.6 for vegetation pixels selection combining the ground observation NDVIs and GOCI pixel NDVI covering the samples, thus, the coverage of the pixel is guaranteed. This explanation is supplemented in section 2.5 of the paper. The NDVI threshold of other phenological phases need further experiment and analysis.

References

  1. Lin, Y.; Shen, H.; Tian, Q.; Gu, X.; Yang, R.; Qiao, B. Mechanisms underlying diurnal variations in the canopy spectral reflectance of winter wheat in the jointing stage. Sci. 2020, 118, 1401–1406, doi:10.18520/cs/v118/i9/1401-1406.

Round 2

Reviewer 1 Report

All my comments have been addressed.